# Synergistic Effect between Zr-MOF and Phosphomolybdic Acid with the Promotion of TiF4 Template

**DOI:** 10.3390/molecules25204673

**Published:** 2020-10-13

**Authors:** Zhu Ding, Xiao Min Zhang, Xue Chang, Shuo Wang, Dan-Hong Wang, Ming Hui Zhang, Tian Hao Zhang

**Affiliations:** 1TKL of Metal and Molecule Based Material Chemistry, National Institute for Advanced Materials, School of Materials Science and Engineering, Nankai University, Tianjin 300350, China; dingzhu91@mail.nankai.edu.cn (Z.D.); zhang_xiaomin163@163.com (X.M.Z.); changxuenku@126.com (X.C.); wshuo@mail.nankai.edu.cn (S.W.); 2School of Physics, Tianjin Key Laboratory of Photonics Materials and Technology for Information Science, Nankai University, Tianjin 300350, China; 3Key Laboratory of Advanced Energy Materials Chemistry (Ministry of Education), College of Chemistry, Nankai University, Tianjin 300071, China; zhangmh@nankai.edu.cn

**Keywords:** MOF materials, Raman spectroscopy, Zr-MOF, SMSI, TiF_4_-PU, oxidative desulfurization

## Abstract

Metal-Organic Framework (MOF) materials are often modified or functionalized, and then the crystal size and morphology of MOF materials are changed. In the process of preparing UiO-66 confined phosphomolybdic acid (PMA) composites (PU), the TiF_4_-modified PU (PMA + UiO-66) composite catalyst (TiF_4_-PU) was successfully synthesized by adding titanium tetrafluoride, and the catalytic desulfurization activity was excellent. Similarly, the reaction mechanism was investigated by means of infrared spectroscopy, Raman spectroscopy, XPS, and UV/Vis spectroscopy. The results show that the addition of TiF_4_ not only changes the appearance and color of the catalyst, but also changes the valence distribution of the elements in the catalyst. The number of oxygen vacancies in the MOF increases due to the addition of TiF_4_, and more electrons are transferred from the Zr-MOF to PMA to form more Mo^5+^, which improved the performance of oxidative desulfurization in comparison. Thus, a stronger strong metal-support interaction (SMSI) effect is observed for TiF_4_-modified PU catalysts. In addition, the quenching experiment of free radicals shows that ·OH radical is the main active substance in the oxidative desulfurization reaction over TiF_4_-PU catalyst.

## 1. Introduction

Petroleum, as a major source of the world’s energy, consists of various sulfur compounds; thus, the combustion can emit amounts of sulfur oxides, eventually causing acid rain [1]. To solve this problem, different techniques are applied to the desulfurization of crude oil [2]. Among these desulfurization approaches, oxidative desulfurization (ODS) is a green method for the removal of sulfur compounds from crude oil [3].

Polyoxometalates (POMs) are composed of heteroatoms (such as P, Si, Fe, Co, etc.) and polyatoms (such as Mo, W, V, Nb, Ta, etc.) according to certain structures [4]. POMs show not only acidity but also redox ability, and act as new green multifunctional catalysts with good structural stabilities [5]. The superiority of the structural certainty behaves as follows: (1) POMs involve the main structural characteristics of both coordination complexes and metal oxides. (2) POMs show the storage and transfer abilities of electrons and protons [6]. (3) Different heteroatoms and polyatoms result in different acidities and redox abilities, which make the catalytic performances controllable and are beneficial to the design of catalysts [7]. Various kinds of POMs, phosphomolybdic acid (PMA with a Keggin structure H_3_PMo_12_O_40_) in particular, have been reported to be efficient catalysts for ODS reaction [8]. Four types of oxygen atoms exist in the Keggin structure (Scheme 1a)**.** Tetrahedral oxygen (O_a_) is coordinated with heteroatom P, bridging oxygen (O_b_) is connected with different trimetallic clusters (Mo_3_O_10_), bridging oxygen (O_c_) is in the same trimetallic cluster (Mo_3_O_10_), and terminal oxygen (O_d_) has a high mobility [9]. O_d_ can be easily lost to create oxygen vacancy, which is considered as the active site as we previously reported [10].

Due to that non-pore and high solubility limiting the use of POMs, Metal-Organic Frameworks (MOFs) were selected as porous supports or host frameworks to encapsulate active POMs for developing effective ODS catalysts with a high recyclability [10,11,12,13,14]. We have also reported that the PMA/UiO-66 composite shows an enhanced ODS activity compared with pure PMA or pure UiO-66. In UiO-66, Zr_6_O_4_(OH)_4_ octahedral nucleus with an eight-coordinated zirconium atom was bridged by carboxylate (-CO_2_) to form a Zr_6_O_4_(OH)_4_(CO_2_)_12_ cluster (Scheme 1b). The most interesting thing is the reducibility of Zr-MOF. The loss of lattice oxygen occurs to form oxygen vacancy, with electrons trapped in when the UiO-66 is treated in N_2_ at a high temperature, as we suggested [10].

Strong metal-support interactions (SMSI) significantly affect the electronic properties of the active components of the catalysts and enhance the catalytic performance [15,16]. When the metal is loaded on the reducible metal oxide support, the reduction of the support at a high temperature leads to a decrease in the H_2_ chemical adsorption in the metal, which is due to the strong interaction between the reducible support and the metal, and the support transfers some electrons to the metal, thus decreasing the chemical adsorption capacity of H_2_ [17,18]. The existence degree of SMSI is related to the reducibility of the support in the reduction process—that is to say, the easier the reduction of the metal oxide supports, the more significant the effects of SMSI [19]. Our previous study [10] revealed that Mo^5+^ and oxygen vacancy can be more easily formed on PMA/UiO-66 composites than on pure PMA or pure UiO-66, indicating that the SMSI effect between PMA and UiO-66 occurred [20].

Templates generally play structure-oriented or ion-oriented roles in the synthesis of MOF materials [21,22,23]. In our previous studies, fluoride has also been used to adjust the morphology of materials, and has obtained the relatively good results [24]. Herein, we find an appropriate template (TiF_4_) to synthesize highly efficient Zr-MOF catalysts with a unique sphere structure by the one-pot method (Scheme 2). Specifically, PMA is encapsulated within UiO-66 to construct a “ship-in-a-bottle” structure (denoted as TiF_4_-PU) [25,26,27]. In this construct, the SMSI effect between the PMA and Zr secondary building units (SBUs) occurred to form heteropoly blue (PMo^VI^_4_Mo^V^_8_O_40_)^11−^ with a Keggin structure [28]. Furthermore, the amount of Mo^5+^ is higher for TiF_4_-PU-200 (Mo^6+^/Mo^5+^ = 4/8 = 0.5) than for PU-200 (Mo^6+^/Mo^5+^ = 8/4 = 2) without a TiF_4_ template. It is noteworthy that TiF_4_-PU shows an obviously enhanced catalytic activity in comparison to the PU catalyst. We believe that, by employing this MOF confinement approach, many kinds of heteropoly blues for various POMs can be generated and confined in MOFs by the TiF_4_ template and expected to show surprising performances for catalysis.

## 2. Materials and Methods

### 2.1. Synthesis of TiF_4_-PU Catalysts

Phosphomolybdic acid was purchased from Meryer, 1,4-benzenedicarboxylic acid and titanium tetrafluoride were purchased from Aladdin, *N*,*N* dimethylformamide(DMF) was purchased from FoChen, zirconium tetrachloride and acetic acid were purchased from Kermel, deionized (DI) water were purchased from ConCord. All chemical reagents are A.R grade and are not further purified after purchase from reagent company.

Phosphomolybdic acid (PMA 23.6 mg, 0.013 mmol), 1,4-benzenedicarboxylic acid (H_2_BDC 83.5 mg, 0.5 mmol), zirconium tetrachloride (ZrCl_4_ 58.25 mg, 0.25 mmol), and titanium tetrafluoride (TiF_4_ 31.6 mg, 0.25 mmol) were dissolved in *N*,*N*-dimethylformamide (DMF 30 mL), including acetic acid (HAc 3.6 mL). After sonication for 30 min, the solution was transferred to a stainless steel autoclave with a polytetrafluoroethylene lining, then it was sealed and kept at 120 °C muffled for 24 h; it was then cooled to room temperature, the solution was centrifugated, the precipitate was washed by DMF and methanol, and it was dried at a temperature of 80 °C. At last, the precursor was activated at different temperatures (200–500 °C) in a nitrogen atmosphere with a tubular resistance furnace for 2 h. TiF_4_-PU-T catalysts were obtained (T represents the activation temperature).

### 2.2. Characterization of Catalysts

X-ray diffraction (XRD) patterns were recorded in the range of 5–80° using a Rigaku MiniFlex 600 with Cu Kα radiation (λ = 0.154178 nm) at 298 K. The BET (Brunner−Emmet−Teller measurements) results of the catalysts were obtained by an ASAP 2460 Micromeritics at 77 K after degassing for 24 h under a vacuum at 60 °C. The morphologies of the samples were analyzed with a JSM-7800F scanning electron microscope (SEM). High-resolution TEM (HRTEM) images were tested using JEM-2800 microscopy. A Bruker TENSOR-37 was used to measure the Fourier transform infrared spectra (FTIR) at 273 K. The XPS results were collected by an ESCALAB 250X-ray from ThermoScientific. Thermo-gravimetric analysis (TG) was conducted by the Thermo plus EVO_2_TG8121 equipment. UV/Vis measurements of the samples were carried out with a TU-1950 PERSEE. A temperature-programmed reduction (H_2_-TPR) analysis was carried out with the HuasiFD-2000 thermal conductivity detector (TCD). Then, 50 mg catalyst was pretreated in an Ar flow (30 mL/min) at 120 °C for 1 h. The reducing gas (5% H_2_ in He) then replaced helium at the same flow rate. The temperature of the reactor increased linearly from 30 to 400 °C at a rate of 5 °C/min.

### 2.3. ODS Reaction Procedure

The preparation of the simulated oil needed in the ODS reaction used decalin as the solvent and dibenzothiophene (DBT) as the solute to form a 500 ppm solution. Tert-butyl hydroperoxide (TBHP) with an O/S molar ratio of 3 was then added to the simulated oil. The solution was stirred evenly as a standby. The specific operation steps of the ODS reaction were as follows: 10 g of 500 ppm solution was weighed and transferred to a 50 mL round bottom flask, which was sat in a heating sleeve with an electromagnetic stirrer. At a certain speed, the temperature was raised to the target temperature of 80 °C verified by a thermocouple. For the use of gas chromatography, when the temperature of the injector, column, and detector rose to the settled temperature (250, 220, and 250 °C, respectively), 0.5 μL of the prepared solution was injected with a syringe. The number of injections was generally 3–5 times. The DBT content obtained was basically consistent with 500 ppm. That is to say, the standard tested by the gas chromatograph is accurate. Subsequently, 50 mg of catalyst was weighed and added to the 500 ppm solution in the round-bottom flask. When the catalyst was added, the recording time was started and set to zero. The sampling interval was 10 min. The DBT content analyzed by the GC 2060 gas chromatograph was recorded and the conversion of DBT under the existence of catalyst was calculated.

To test the cycling performance of the catalyst, the steps were as follows: we centrifuged the reacted solution with methanol repeatedly, washed and immersed the recycling catalyst in methanol for 1–2 days to remove the adsorbed product of sulfone, and then centrifuged and dried it to obtain the recycling catalyst.

## 3. Results and Discussion

### 3.1. Characterization of TiF_4_-PU-T

We have previously reported the synthesis of PU-200 by confining PMA with UiO-66 using the one-pot method [10]. The difference for the synthesis of TiF_4_-PU is the addition of TiF_4_ to the solution using the same one-pot method. The XRD patterns of TiF_4_-PU-T are shown in Figure 1a. When the calcination temperature is below 300 °C, the XRD patterns of TiF_4_-PU are consistent with those of UiO-66 and PU-200. No other diffraction peaks are found, indicating that the PMA is well dispersed in UiO-66. Above 300 °C, the structure of UiO-66 collapsed to form ZrO_2_. This result is consistent with TG result of the TiF_4_-PU precursor, as shown in Appendix A. The thermal stability of TiF_4_-PU is worse than that of PU (which can maintain a UiO-66 structure up to 400 °C) [10], implying the easy loss of the lattice oxygen of Zr-MOF in an inert atmosphere for TiF_4_-PU.

The FT-IR spectra are shown in Figure 1b; the tensile vibration peak of Mo = O_d_ (958 cm^−1^) in TiF_4_-PU (up to 300 °C) is consistent with that in PMA, indicating that the PMA in the TiF_4_-PU catalysts maintains a Keggin structure up to 300 °C. Moreover, the FT-IR spectra also confirm that the UiO-66 structure keeps in TiF_4_-PU up to 300 °C due to the weakening of the UiO-66 peaks above 300 °C, which is consistent with the XRD results.

From the images of the TiF_4_-PU catalysts shown in Figure 1c, it is very interesting that dark blue colors are observed for TiF_4_-PU catalysts up to 200 °C compared with the light blue color for the PU-200 catalyst. In our previous study [10], we have found when PMA is confined in UiO-66 to obtain PU-200 catalyst, heteropoly blue (PMo^VI^_8_Mo^V^_4_O_40_)^7−^ with a Keggin structure is formed. Thus, also according to the FT-IR results, we can infer that more Mo^5+^ with a Keggin structure is formed in the TiF_4_-PU catalysts than in the PU-200 catalyst, which causes darker blue colors for TiF_4_-PU catalysts than for the PU-200 catalyst.

The BET results are shown in Appendix A and Table 1. We can clearly see that, up to 200 °C, the surface areas for the TiF_4_-PU catalysts changed a little compared with PU-200. Meanwhile, above 200 °C the surface area and pore volume for the catalysts decreased a lot due to the collapse of the UiO-66 structure. Thus, we can suggest that the introduction of TiF_4_ does not affect the pore structure of the TiF_4_-PU catalysts up to 200 °C, compared with that of PU-200.

The SEM and TEM images for the TiF_4_-PU catalysts are shown in Appendix A and Figure 2, respectively. It is very interesting that the morphology of the TiF_4_-PU catalysts can be changed by the introduction of TiF_4_ compared with PU-200. We tried TiF4 with different concentrations, and the final morphologies obtained were all spherical. We also tried to add TiF4 without PMA; the morphology of MOF is still spherical, and the change in the morphology of MOF is caused by TiF4. As seen in Appendix A, UiO-66 and PU-200 show the morphology of a regular octahedron, while the TiF_4_-PU catalysts show the morphology of a sphere. It can be inferred that the introduction of F^−^ can etch the corners of the octahedron UiO-66 to form the sphere. As seen in Figure 2, the TEM element mapping images for TiF_4_-PU-200 show that the Mo and P elements are highly dispersed in the whole spheres, indicating that PMA is highly dispersed in UiO-66. Moreover, the TEM-EDS results show that few Ti and F elements are detected in the whole spheres, indicating that TiF_4_ acts only as a template in the preparation procedure for TiF_4_-PU catalyst and does not participate in the composition of the catalyst.

We performed XPS analysis to obtain the electronic structure of the TiF_4_-PU catalysts in Figure 3. The binding energies (BE) of Zr_3d_ (Figure 3c) for he PU and TiF_4_-PU catalysts increased compared with pure UiO-66, and the BE of P_2p_ (Figure 3d) for PU and TiF_4_-PU catalysts decreased compared with pure PMA. This fact strongly indicates that strong metal-support interactions (SMSI) occurred between PMA and UiO-66 in the TiF_4_-PU catalysts, as we obtained before for the PU catalysts [10]. According to the BE of Mo_3d_ (Figure 3a) for the PU and TiF_4_-PU catalysts, Mo^5+^ was found to be formed compared with pure PMA, and the molar ratio of Mo^6+^/Mo^5+^ in the PU and TiF_4_-PU catalysts is calculated from Appendix A and shown in Table 2. Zr-MOF easily loses oxygen atoms in the lattice to form oxygen vacancies that trap electrons. The energy state of the oxygen vacancy is very close to the conduction band position of Zr-MOF. Thus, the electrons captured by the oxygen vacancy can be easily excited to the Zr-MOF conduction band. The electrons then transfer from the Zr-MOF conduction band to the PMA conduction band to form the band transition, eventually leading to the formation of Mo^5+^ [10]. V_O_^X^ is the oxygen in the lattice, and O_O_^••^ is the oxygen vacancy in Equation (1). Equations (2) and (3) shows the changes in the Mo^5+^ and Mo^6+^ contents. The TiF_4_-PU-200 catalyst shows the highest Mo^5+^ content. It is very interesting that (PMo^VI^_8_Mo^V^_4_O_40_)^7−^ are formed in PU-200 as Equation (2), and (PMo^VI^_4_Mo^V^_8_O_40_)^11−^ are formed in TiF_4_-PU-200 as Equation (3). According to the BE of O_1s_ (Figure 3b) for PU and TiF_4_-PU catalysts, oxygen vacancy (532 eV) are found to be formed compared with pure UiO-66 or PMA. Further, compared with PU-200, the TiF_4_-PU-200 composite tends to lose lattice oxygen in the UiO-66 framework and form oxygen vacancies with more free electrons, as shown in Equation (1), and the oxygen vacancy content of TiF_4_-PU-200 is the highest (Appendix A). The fact is that more Mo^5+^ and oxygen vacancy are formed in TiF_4_-PU-200 than in PU-200, which highlights the function of TiF_4_ as a promoter. The BE of Mo spectrum decreased, which implies that Mo was reduced inherently. On the basis of these observations, the molar ratio of Mo^6+^/Mo^5+^ in TiF_4_-PU-200 catalyst can account for 2, while in the PU-200 composite it is 0.5, which indirectly proves the necessity and importance of TiF_4_ as the regulator to reduce Mo^6+^. It is supposed that DMF can reduce Ti^4+^ to Ti^3+^ easily, and thus Ti^3+^ can reduce Mo^6+^ to form more Mo^5+^. According to all the above results, the easier the reduction of the TiF_4_-PU catalyst, the more significant the effects of SMSI, and we can say that stronger SMSI effects occurred in the TiF_4_-PU-200 catalyst than in the PU-200.
(1)$$O_{O}^{X}\to V_{O}^{\cdot \cdot }+2e+\frac{1}{2}O_{2}$$
(2)$${\left({{\mathrm{PM}}_{O}}^{VI}{}_{12}O_{40}\right)}^{3-}+4e\overset{DMF}{\to }{\left({{\mathrm{PM}}_{O}}^{VI}{}_{8}{M_{O}}^{V}{}_{4}O_{40}\right)}^{7-}$$
(3)$${\left({{\mathrm{PM}}_{O}}^{VI}{}_{12}O_{40}\right)}^{3-}+8e\overset{\frac{DMF}{T_{i}F_{4}}}{\to }{\left({{\mathrm{PM}}_{O}}^{VI}{}_{4}{M_{O}}^{V}{}_{8}O_{40}\right)}^{11-}$$

To understand the origin of the strong SMSI effects for TiF_4_-PU-200, the UV/Vis diffuse reflection spectra (a) and TPR results are obtained for the catalysts (Figure 4). As can be seen from Figure 4a, the precursor and TiF_4_-PU-200 exhibits absorption edge at 358 cm^−1^ and 376 cm^−1^, while PU-200 exhibits an absorption edge at 327 cm^−1^. Using Eg = hc/λ = 1240/λ (h is Planck constant, c is light speed), the band gap (Eg) of semiconductor can be calculated from the wavelength (λ) of absorption edge (the wavelength is the intercept of X axis). Thus, the energy gap for PU-200, precursor and TiF_4_-PU-200 are calculated to be 3.79, 3.46 and 3.29 eV, respectively. The narrower energy gap for TiF_4_-PU-200 can be explained by the formation of more oxygen vacancies in TiF_4_-PU-200 than that in PU-200, which results in the introduction of defect band (V_O_^··^) under the conduction band (CB) of UiO-66. The adsorption below 325 nm can be assigned to the electron transition from oxygen (O) to metal (M), the adsorption in the range of 325–400 nm is attributed to the resonance of Mo^5+^, and the absorption band in the ultraviolet absorption region of 600–800 cm^−1^ is caused by the d-d electron transition of heteropoly blue containing Mo^5+^. With the influence of regulator TiF_4_, the Mo^5+^ signal significantly increased. From the TPR results shown in Figure 4b, the peak of hydrogen consumption for the TiF_4_-PU-200 composite (264 °C) shifts to a lower temperature than that for PU-200 (245 °C)**.** This peak is attributed to the loss of lattice oxygen in UiO-66, as we have reported before [10]. Thus, TiF_4_-PU-200 is easier to reduce to form more oxygen vacancies than PU-200. In summary, according to UV/Vis DRS and TPR results, TiF_4_-PU-200 possesses more Mo^5+^ content and more oxygen vacancies than PU-200.

### 3.2. ODS Activity

As obtained above, more Mo^5+^ and oxygen vacancies are generated in TiF_4_-PU-200 than in PU-200. This attracted us to systematically study the ODS activity and ODS mechanism to confirm the difference in the SMSI effect between TiF_4_-PU-200 and PU-200. The oxidative desulfurization performance of TiF_4_-PU composites was tested by the oxidation of DBT using TBHP as the oxidant. On the basis of previous studies, we determined that the ODS reaction temperature was 80 °C and the reaction test time was 2 h. We explored the effect of calcination temperature on the catalysts (Figure 5a). The ICP results of the PU and TiF_4_-PU catalysts are shown in Appendix A; the Mo contents for the PU-200 and TiF_4_-PU-200 catalysts are almost the same (8 wt.%), and TiF_4_-PU-200 shows the highest ODS activity among the TiF_4_-PU catalysts, which can be attributed to the highest Mo^5+^ content. TiF_4_-PU-200 shows a higher ODS activity than PU-200, which can be attributed to the higher Mo^5+^ content, as we obtained above, and heteropoly blue (PMo^VI^_4_Mo^V^_8_O_40_)^11−^ in formed in TiF_4_-PU-200 and (PMo^VI^_8_Mo^V^_4_O_40_)^7−^ is formed in PU-200. In Figure 5b, the ODS efficiency drops completely in the presence of DMSO (quencher of ·OH). This result provides evidence that ·OH is an important reactive species in the ODS process. It is worth emphasizing that ·OH possesses a high oxidability to oxidize DBT.

In addition, we also investigated the recyclability of the TiF_4_-PU-200 composite. The results show that the catalyst maintained the characteristic XRD peak of XRD and still maintained a good catalytic activity after five cycles, as shown in Figure 6a. Figure 6b shows that the catalytic materials can be reused. After five cycles of testing, they still have a good catalytic activity, and the conversion rate does not decrease basically. This result suggests that the TiF_4_-PU-200 composite possesses a good structural stability and ODS activity stability.

### 3.3. ODS Reaction Mechanism

We have studied the catalytic properties of TiF_4_-PU-200 compared with PU-200. The detailed mechanism is described as shown in Scheme 3. The energy of conduction band (*E*_CB_) of UiO-66 is reported to be −0.09 eV versus NHE, and the *E*_CB_ of PMA is reported to be 0.65 eV versus NHE [30,31]. The lattice oxygen of Zr-MOF is easily lost to form oxygen vacancies with the electrons trapped in. The defect band (V_O_^··^) is very close to the conduction band (CB) of UiO-66. Subsequently, the abundant electrons are excited to the conduction band of UiO-66, then transferred to the conduction band of PMA, forming a band–band transition, which leads to forming more Mo^5+^ and a high electron density on the Mo atom, as reported in our previous work. Thus, the fermi level (EF) for PMA increased. According to the DFT calculation, TBHP adsorbed on Mo(V) more easily than on Mo(VI) to form ·OH active substances, which can explain the synergistic effect. The hydroxyl radical that forms has a very strong oxidizing ability. The high electron density makes it easier for Mo^5+^ to combine with TBHP (tert-butyl hydrogen peroxide) to form ·OH active species, which has an important synergistic effect on the ODS mechanism, which can oxidize DBT into DBTO_2_. In all, the truth was that the TiF_4_ reagent leads to more electron transfer from UiO-66 to PMA as the result of more Mo^5+^. To our knowledge, this is the first finding that when TiF_4_ is added to the synthetic solution as the modifier, it can promote more electron transfer from UiO-66 to PMA compared with PU-200.

## 4. Conclusions

In summary, TiF_4_ promoted the synergistic effect (SMSI) between PMA and UiO-66, including electron transfer, and thus achieved a better oxidative desulfurization activity. From the SEM and TEM analyses, we believe that the regulator of TiF_4_ directly affects the morphology of the composite catalyst to obtain spheres. The XPS, UV/Vis DRS, and TPR results suggest that TiF_4_ promoted the formation of more oxygen vacancies in UiO-66, inducing more Mo^5+^ formation and speeding up the ODS reaction. The use of a regulator greatly decreased the reduction temperature of the catalyst. Heteropoly blue (PMo^VI^_8_Mo^V^_4_O_40_)^7−^ with a Keggin structure is formed in PU-200 and (PMo^VI^_4_Mo^V^_8_O_40_)^11−^ is formed in TiF_4_-PU-200. At the same time, this is the first time that an inorganic template has been used in the synthesis of MOF composites, which provides a new direction for the synthesis of MOF composite catalysts in the future.

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
