# Peer review of "Synergistic Effect between Zr-MOF and Phosphomolybdic Acid with the Promotion of TiF4 Template"

_molecules, 2020, doi:10.3390/molecules25204673_

Round 1

Reviewer 1 Report

Comments to the author:

In this manuscript, the authors presented the modification of UiO66 MOF with phosphomolybdic acid (PMA), an efficient catalysts for oxidative desulfurization ODS reaction and TiF4 which promoted the synergistic effect (SMSI) between PMA and UiO-66 including electron transfer. The figure of merit of this work is the novel use of TiF4, wich provided to the MOF a better oxidative desulfurization activity most probably related to the more Mo5+ formation. The authors compared the characterization of the MOF with and without TiF4 and optimized the temperature of fabrication. Overall, I would recommend the publication of this contribution because this is the first time that inorganic template is used in the synthesis of MOF composites, which provides a new direction for the synthesis of MOF composite catalysts in the future.

The following are some questions and suggestions for improving their work:

Major issues:

  1. The authors should comment on other works where MOFs and TiF4 are used together.
  2. The authors have tried to prepare the MOF at different temperatures but did they try to use different amounts of TiF4? If so, was the final shape of the MOF spherical?
  3. If the authors perform the same fabrication without using PMA, is the final shape of the MOF spherical as well?
  4. When the authors show the ODS activities of TiF4-PU-200 and PU-200 the activity is quite similar, the only difference is the time to reach 100%conversion which are 30 and 50 min. So in the opinion of the authors, can this time decrease can be considered as fundamental?

Minor issues:

  1. What the authors refer to precursor TiF4-PU? Is that room temperature?
  2. Figure S4 and S5 have no letters. They should be added.
  3. Figure 4. Absorbance has no units, no arbitrary units.
  4. The axis of Figure 5 should be modified in order to clearly show the data.

Author Response

Thank you very much for your suggestions, which are very helpful to us. We have answered and modified them according to your suggestions and questions, which are as follows:

Major issues:

1.The authors should comment on other works where MOFs and TiF4 are used together.

Thank you for the reviewer’s important suggestion.As far as we know, no one has handled MOF with TiF4 before us, so it is the first attempt to modify MOF with this method.

2.The authors have tried to prepare the MOF at different temperatures but did they try to use different amounts of TiF4? If so, was the final shape of the MOF spherical?

Thank you for the reviewer’s meticulous work on our manuscript.We tried TiF4 with different concentrations, and the final morphologies obtained were all spherical.

3.If the authors perform the same fabrication without using PMA, is the final shape of the MOF spherical as well?

Thank you for the reviewer’s conscientious work. We have tried to add TiF4 without PMA, the morphology of MOF is still spherical, and the change of the morphology of MOF is caused by TiF4

4.When the authors show the ODS activities of TiF4-PU-200 and PU-200 the activity is quite similar, the only difference is the time to reach 100%conversion which are 30 and 50 min. So in the opinion of the authors, can this time decrease can be considered as fundamental?

Thank you for the reviewer’s meticulous work on our manuscript. As can be seen from Fig 5a, TiF4-PU-200 and PU-200 showed obvious differences in activity 10 minutes after the reaction. At last, when the conversion rate reached 100%, TiF4-PU-200 was more than 20 minutes earlier than PU-200, so we believed that there was a big difference in activity between the two catalysts.

Minor issues:

1.What the authors refer to precursor TiF4-PU? Is that room temperature?

Thank you for the reviewer’s contributions to our work. The precursor TiF4-PU means that there is no pre-calcined sample.That is room temperature.

2.Figure S4 and S5 have no letters. They should be added.

Thank you for the reviewer’s important suggestion, which is very helpful to us, and we have modified it.Every figure add the letters.

3.Figure 4. Absorbance has no units, no arbitrary units.

Thank you very much for your valuable advice, which is very helpful to us, and we have modified it.

4.The axis of Figure 5 should be modified in order to clearly show the data.

Thank you for the reviewer’s meticulous work on our manuscript, which is very helpful to us, and we have modified it.

Reviewer 2 Report

Dear Authors and Editors,

Here are some comments and questions concerning the considered article:

  1. The title of the article looks more like the thesis in the Conclusions section, so I think it should be reformulated.
  2. It is better to avoid abbreviations in the title and Abstract section as it is in general difficult for the readers to understand their meaning. Lots of abbreviations in the discussion section complicate the understanding of experimental results, so in my opinion, their quantity should be decreased. The description of the experimental data in the text in some parts of the manuscript is rather brief and is difficult to follow.
  3. Lines 145 and 146 concerning previously mentioned one-pot synthesis do not contain the corresponding reference, maybe it is missing.
  4. In section 3.1 rather complicated figure 1 is located above the text in which it is discussed and it is not a good choice as the reader has to scroll for the description of the data on the figures.
  5. Equations 1-3 should be discussed in the text in a more detailed way, the notations are difficult to understand.
  6. 6 caption should be more detailed with the specification of a) and b) parts. B) part of the figure is not completely clear: what is the meaning of a histogram? To show that all 5 cycles the conversion is close to 100% or what? What is the meaning of different colors of the columns?
  7. Scheme 3 should be discussed in a more comprehensive and detailed way, it is difficult to follow the author’s text and compare it with the illustration on scheme 3 with not enough provided details and notation description.

Author Response

Thank you very much for your suggestions, which are very helpful to us. We have answered and modified them according to your suggestions and questions, which are as follows:

1.The title of the article looks more like the thesis in the Conclusions section, so I think it should be reformulated.

Thank you very much for your valuable advice, which is very helpful to us, and we have modified it. The title of the manuscript is changed toSynergistic effect between Zr-MOF and phosphomolybdic acid with the promotion of TiF4 template”.

2.It is better to avoid abbreviations in the title and Abstract section as it is in general difficult for the readers to understand their meaning. Lots of abbreviations in the discussion section complicate the understanding of experimental results, so in my opinion, their quantity should be decreased. The description of the experimental data in the text in some parts of the manuscript is rather brief and is difficult to follow.

Thank you for the reviewer’s good question, which is very helpful to us. We have revised the title and abstract to minimize the use of abbreviations and annotate the abbreviations to make them easier for readers to understand. The changes have been highlighted.

Abstract:MOF materials are often modified or functionalized, and then the crystal size and morphology of MOF materials are changed. In the process of preparing UiO-66 confined PMA(phosphomolybdic acid) composites(PU), the TiF4 modified PU(PMA+UiO-66) composite catalyst (TiF4-PU) was successfully synthesized by adding titanium tetrafluoride and the catalytic desulfurization activity was excellent. Similarly, the reaction mechanism was investigated by means of Infrared spectroscopy, Raman spectroscopy and XPS, UV-vis spectroscopy. The results show that the addition of TiF4 not only changes the appearance color of the catalyst, but also changes the valence distribution of elements in the catalyst. The number of oxygen vacancies in MOF increases due to the addition of TiF4, more electrons are transferred from Zr-MOF to PMA to form more Mo5+, which improved the performance of oxidative desulfurization compared. Thus, stronger SMSI(strong metal-support interaction) effect is observed for TiF4-modified PU catalysts. In addition, the quenching experiment of free radicals shows that ·OH radical is the main active substance in the oxidative desulfurization reaction over TiF4-PU catalyst.

3.Lines 145 and 146 concerning previously mentioned one-pot synthesis do not contain the corresponding reference, maybe it is missing.

Thank you for the reviewer’s contributions to our work. As the reviewer suggested, we have modified it. The specific additions are as follows:

We have previously reported the synthesis of PU-200 by confining PMA with UiO-66 using one pot method[10].The difference for the synthesis of TiF4-PU is the addition of TiF4 to the solution using the same one pot method. XRD patterns of TiF4-PU-T are shown in Fig. 1a.

4.In section 3.1 rather complicated figure 1 is located above the text in which it is discussed and it is not a good choice as the reader has to scroll for the description of the data on the figures.

Thank you for the reviewer’s conscientious work. As the reviewer suggested, we have modified it. We have placed the complex figure 1 below the text for easy reading.

5.Equations 1-3 should be discussed in the text in a more detailed way, the notations are difficult to understand.

Thank you for the reviewer’s meticulous work on our manuscript, We further explained the equations and some notations in the equations are elaborated.

Zr-MOF easily loses the oxygen atoms in the lattice to form oxygen vacancies that trap electrons.The energy state of the oxygen vacancy is very close to the conduction band position of Zr-MOF.Thus, the electrons captured by the oxygen vacancy can be easily excited to the Zr-MOF conduction band.The electrons then transfer from the Zr-MOF conduction band to the PMA conduction band to form the band transition, eventually leading to the formation of Mo5+[10].OOâ…¹ is oxygen in the lattice and VO•• is Oxygen vacancy in Equ. 1.Equ.2 and Equ.3 shows the changes of Mo5+ and Mo6+ contents.

6.6 caption should be more detailed with the specification of a) and b) parts. B) part of the figure is not completely clear: what is the meaning of a histogram? To show that all 5 cycles the conversion is close to 100% or what? What is the meaning of different colors of the columns?

Thank you very much for your valuable advice, which is very helpful to us. Fig. 6b shows that the catalytic materials can be reused. After five cycles of testing, they still have good catalytic activity, and the conversion rate does not decrease basically. We also added explanations to the manuscript and modified the diagram.

In addition, we also investigated the recyclability of the TiF4-PU-200 composite. The result show that the catalyst maintained the characteristic XRD peak of XRD and still maintained good catalytic activity after five cycles as shown in Fig. 6a.Fig. 6b shows that the catalytic materials can be reused. After five cycles of testing, they still have good catalytic activity, and the conversion rate does not decrease basically. This result suggests that TiF4-PU-200 composite possesses good structure stability and ODS activity stability.

7.Scheme 3 should be discussed in a more comprehensive and detailed way, it is difficult to follow the author’s text and compare it with the illustration on scheme 3 with not enough provided details and notation description.

Thank you for the reviewer’s important suggestion, we have described scheme 3 further and explained some abbreviations to make it easier to understand,and also provides some details.

We have studied the catalytic properties of TiF4-PU-200 compared with PU-200. Detailed mechanism is described as shown in Scheme 3. The energy of conduction band (ECB) of UiO-66 is reported to be -0.09 eV versus NHE, and the ECB of PMA is reported to be 0.65 eV versus NHE.[30,31] Lattice oxygen of Zr-MOF is easily lost to form oxygen vacancies with electrons trapped in. The defect band (VO..) is very close the conduction band (CB) of UiO-66. Subsequently, the abundant electrons are excited to the conduction band of UiO-66, then transferred to the conduction band of PMA forming a band - band transition, which leads to form more Mo5+ and high electron density on Mo atom as reported in our previous work. Thus the fermi level (EF) for PMA increased.According to DFT calculation, TBHP adsorbed on Mo(V) more easily than on Mo(VI) to form ·OH active substances, which can explain the synergistic effect.The hydroxyl radical that forms has very strong oxidizing ability. The high electron density makes it easier for Mo5+ to combine with TBHP(Tert-butyl hydrogen peroxide) to form ·OH active species that has an important synergistic effect on ODS mechanism, which can oxidize DBT into DBTO2. In all, the truth was that the TiF4 reagent leads to more electron transfer from UiO-66 to PMA as the result of more Mo5+. To our knowledge, this is the first finding that when TiF4 is added to the synthetic solution as the modifier, it can promote more electron transfer from UiO-66 to PMA compared with PU-200.

Reviewer 3 Report

Wang and Zhang et al. report on the preparation of new oxidative desulfurization (ODS) catalytic systems based on UiO-66 containing phosphomolybdic acid (PMA). TiF4 was used for this preparation to get new TiF4-PU catalyst. Previously, they reported PU-200 without using TiF4. They claimed that TiF4-PU-200 showed better ODS activity than PU-200 due to increase of Mo5+ and O vacancies. Overall, the research topic, experiments, and the analyses of results seem to be fine for the publication. Nonetheless, the authors need to address the following issues before final decision:

(1) Does "PU" mean "PMA + UiO-66"? If so, it can be more clearly indicated in early part of text. Because PU sometimes stands for polyurethane.

(2) The reasoning for the choice of TiF4 can be clarified in introduction.

(3) Table 1: the unit of BET surface area may be misspelled. Also, the decimal point values for TiF4-300/400/500 can be removed. BET surface area values are not that accurate.

(4) Line 214/215: both eqs. 2 & 3 should be rewritten.

(5) Scheme 3: what is TBHP? Please indicate this in the scheme.

(6) The full names of PMA (phosphomolybdic acid) and SMSI (strong metal-support interaction) should be indicated in the Abstract.

(7) Experimental section: mmol numbers of each chemical should also be indicated along with xx mg.

(8) More Keywords can be added. Now there are only three.

(9) The expression "bottle around ship" can be expressed either in "bottle-around-a-ship" or in "ship-in-a-bottle".

(10) Line 286: the expression of oxidation state of metal ions are a little bit confusing. (PMoVI8MoV4O40)7- can be better than the current (PMo8VIMo4VO40)7-.

Author Response

Thank you very much for your suggestions, which are very helpful to us. We have answered and modified them according to your suggestions and questions, which are as follows:

(1)Does "PU" mean "PMA + UiO-66"? If so, it can be more clearly indicated in early part of text. Because PU sometimes stands for polyurethane.

Thank you for the reviewer’s important suggestion, which is very helpful to us. "PU" means "PMA + UiO-66" and we have indicated in early part of text.

MOF materials are often modified or functionalized, and then the crystal size and morphology of MOF materials are changed. In the process of preparing UiO-66 confined PMA(phosphomolybdic acid) composites(PU), the TiF4 modified PU(PMA+UiO-66) composite catalyst (TiF4-PU) was successfully synthesized by adding titanium tetrafluoride and the catalytic desulfurization activity was excellent.

(2) The reasoning for the choice of TiF4 can be clarified in introduction.

Thank you for the reviewer’s contributions to our work. We have explained why TiF4 was selected in introduction and cited relevant literatures.

Templates generally play structure-oriented or ion-oriented roles in the synthesis of MOFs materials[21-23]. In our previous studies, fluoride has also been used to adjust the morphology of materials, and obtained the relatively good results[24].Herein, we find an appropriate template (TiF4) to synthesize high efficient Zr-MOF catalysts with unique sphere structure by one-pot method (Scheme 2).

(3) Table 1: the unit of BET surface area may be misspelled. Also, the decimal point values for TiF4-300/400/500 can be removed. BET surface area values are not that accurate.

Thank you very much for your valuable advice, which is very helpful to us, and we have modified it. The specific modifications are as follows:

Sample

BET (m²/g)

Pore Volume (cm3/g)

Pore size (Å)

PU-200

992

0.54

-

precursor

858

0.50

5.26

TiF4-PU-100

854

0.47

5.34

TiF4-PU-200

703

0.38

5.29

TiF4-PU-300

331

0.19

4.98

TiF4-PU-400

83

0.07

5.72

TiF4-PU-500

15

0.04

5.74

(4) Line 214/215: both eqs. 2 & 3 should be rewritten.

Thank you very much for your valuable advice, which is very helpful to us, and we have modified it.

(5) Scheme 3: what is TBHP? Please indicate this in the scheme.

Thank you for the reviewer’s good question, We have added a specific explanation of TBHP in the article, as follows:

The high electron density makes it easier for Mo5+ to combine with TBHP(Tert-butyl hydrogen peroxide) to form ·OH active species, which can oxidize DBT into DBTO2. In all, the truth was that the TiF4 reagent leads to more electron transfer from UiO-66 to PMA as the result of more Mo5+.

(6) The full names of PMA (phosphomolybdic acid) and SMSI (strong metal-support interaction) should be indicated in the Abstract.

Thank you for the reviewer’s meticulous work on our manuscript, which is very helpful to us, and we have modified it.

Abstract:MOF materials are often modified or functionalized, and then the crystal size and morphology of MOF materials are changed. In the process of preparing UiO-66 confined PMA(phosphomolybdic acid) composites(PU), the TiF4 modified PU(PMA+UiO-66) composite catalyst (TiF4-PU) was successfully synthesized by adding titanium tetrafluoride and the catalytic desulfurization activity was excellent. Similarly, the reaction mechanism was investigated by means of Infrared spectroscopy, Raman spectroscopy and XPS, UV-vis spectroscopy. The results show that the addition of TiF4 not only changes the appearance color of the catalyst, but also changes the valence distribution of elements in the catalyst. The number of oxygen vacancies in MOF increases due to the addition of TiF4, more electrons are transferred from Zr-MOF to PMA to form more Mo5+, which improved the performance of oxidative desulfurization compared. Thus, stronger SMSI(strong metal-support interaction) effect is observed for TiF4-modified PU catalysts. In addition, the quenching experiment of free radicals shows that ·OH radical is the main active substance in the oxidative desulfurization reaction over TiF4-PU catalyst.

(7) Experimental section: mmol numbers of each chemical should also be indicated along with xx mg.

Thank you very much for your valuable advice, which is very helpful to us, and we have modified it.

Phosphomolybdic acid (PMA 23.6 mg,0.013mmol), 1,4-benzenedicarboxylic acid (H2BDC 83.5mg,0.5mmol), zirconium tetrachloride (ZrCl4 58.25mg,0.25mmol) and titanium tetrafluoride (TiF4 31.6mg,0.25mmol) were dissolved in N,N-dimethylformamide (DMF 30 mL) including acetic acid (HAc 3.6 mL).

(8) More Keywords can be added. Now there are only three.

Thank you for the reviewer’s contributions to our work, which is very helpful to us, and we have modified it.

Keywords: MOF materials;Raman spectroscopy;Zr-MOF;SMSI;TiF4-PU; oxidative desulfurization.

(9) The expression "bottle around ship" can be expressed either in "bottle-around-a-ship" or in "ship-in-a-bottle".

Thank you for the reviewer’s meticulous work on our manuscript, which is very helpful to us, and we have modified it.

(10) Line 286: the expression of oxidation state of metal ions are a little bit confusing. (PMoVI8MoV4O40)7- can be better than the current (PMo8VIMo4VO40)7-.

Thank you for the reviewer’s good question, which is very helpful to us, and we have modified it.

Round 2

Reviewer 2 Report

In general, I agree with the authors' corrections and approve them.

I have some general remarks:

-fig. 1c caption is shifted and badly seen.

-DMF and DMF/TiF4 legend blocks to see equations 2 and 3 under it.

- red color is too bright for the eye on histogram 6b. Maybe it is worth including error bars or specifying in the text the exact decrease in percents which is observed after 5 cycles (it can be seen that the 5th column is slightly lower in height but the scale of the histogram does not allow to estimate the decrease). 

Author Response

Thank you very much for your valuable advice, which is very helpful to us, and we have modified it.
